# Visualization of a multi-turnover Cas9 after product release

Kaitlyn A. Kiernan [1,5] & David W. Taylor [1,2,3,4] ✉

While the most widely used CRISPR-Cas enzyme is the Cas9 endonuclease from *Streptococcus pyogenes* (Cas9), it exhibits single-turnover enzyme kinetics which leads to long residence times on product DNA. This blocks access to DNA repair machinery and acts as a major bottleneck during CRISPR-Cas9 gene editing. Cas9 can eventually be removed from the product by extrinsic factors, such as translocating polymerases, but the mechanisms contributing to Cas9 dissociation following cleavage remain poorly understood. Here, we employ truncated guide RNAs as a strategy to weaken PAM-distal nucleic acid interactions and promote faster enzyme turnover. Using kinetics-guided cryo-EM, we examine the conformational landscape of a multi-turnover Cas9, including the first detailed snapshots of Cas9 dissociating from product DNA. We discovered that while the PAM-distal product dissociates from Cas9 following cleavage, tight binding of the PAM-proximal product directly inhibits re-binding of new targets. Our work provides direct evidence as to why Cas9 acts as a single-turnover enzyme and will guide future Cas9 engineering efforts.

The CRISPR-Cas9 system (Clustered Regularly Interspaced Short Palindromic Repeats-CRISPR associated) from *Streptococcus pyogenes* carries out RNA-guided DNA target recognition and cleavage and has been repurposed for precise genome manipulation[1,2]. Programmed with a single-guide RNA (sgRNA), Cas9 specifically targets and cleaves double-stranded DNA sequences flanked by a protospacer-adjacent motif (PAM). Following DNA cleavage, Cas9 remains stably associated with the dsDNA, limiting product release, and results in single-turnover cleavage kinetics (Fig. 1a)[3]. This persistent product-bound state precludes access to the double-strand break (DSB) in vivo, leads to slow DNA repair, and ultimately dictates repair outcomes[3–6]. While stable association can be beneficial for technologies that require prolonged Cas9 residence times, such as CRISPRi[7–12], this slow dissociation acts as a barrier to achieving efficient gene editing in vivo.

Cas9 is a highly modular enzyme that undergoes large structural transitions during each stage of its reaction pathway, enabling conformational control of Cas9 activity[1,3,13–22]. Cas9 first binds to dsDNA

targets via weak PAM interactions and triggers initial melting through structural distortion of the DNA[18,21–23]. This melting enables Cas9 to probe for complementarity between the target strand (TS) of the DNA and spacer region of the sgRNA, proceeding in a directional manner towards the PAM-distal end. During R-loop propagation, the REC2 and REC3 domains are displaced from the central binding channel to accommodate the sgRNA-TS duplex, and once the R-loop proceeds past 14-bp, REC3 docks onto the PAM-distal region of the duplex[13,15,17,18]. Full accommodation of the R-loop in the central channel creates a kink in the DNA and 'unlocks' the HNH domain, enabling a 34 Å translation and ~180° rotation towards the scissile phosphate at position 3 of the TS[19–21]. DNA unwinding and subsequent R-loop formation is the major energetic and kinetic barrier for Cas9 activation, and once this is complete, HNH and RuvC rapidly cleave the TS and NTS in a sequential, concerted manner[21,24–26].

Cas9 activity being coupled to stable R-loop formation imparts an immense advantage for programmable gene editing and enables Cas9

[1]Department of Molecular Biosciences, University of Texas at Austin, Austin, TX, USA. [2]Institute for Cellular and Molecular Biology, University of Texas at Austin, Austin, TX, USA. [3]Center for Systems and Synthetic Biology, University of Texas at Austin, Austin, TX, USA. [4]Livestrong Cancer Institutes, Dell Medical School, Austin, TX, USA. [5]Present address: Institute of Science and Technology Austria, Am Campus 1, A-3400 Klosterneuburg, Austria. ✉e-mail: dtaylor@utexas.edu

activation when loaded with any sgRNA containing sufficient complementary to a PAM-containing DNA target. However, this stable R-loop formation prevents DNA re-winding following cleavage and leads to extremely slow dissociation rates, preventing release of DSBs and affecting Cas9 efficiency. Long Cas9 residence times have also been implicated in biasing DNA repair outcomes and is an important consideration for in vivo applications[5,6]. Several studies have shown that Cas9 dissociation requires translocating enzyme complexes, such as RNA polymerase, moving replication forks, and chromatin remodelers, to effectively dislodge Cas9 from a DSB[6,27-30]. It has also been shown that Cas9 dissociation occurs more readily if the collision with a translocating enzyme occurs on the PAM-distal side of Cas9[24,26,27], however, there is no unifying mechanism describing Cas9 dissociation to support this.

To address this gap in the field, we use a combination of kinetics and cryo-electron microscopy (cryo-EM) to characterize the mechanisms controlling Cas9 turnover. Here, we carry out kinetic analysis of Cas9 with altered guide-target duplexes and identify a sgRNA truncation that induces multi-turnover cleavage kinetics. Cryo-EM analysis of a multi-turnover complex enabled us to capture snapshots of Cas9 throughout the entire reaction cycle, including direct visualization of Cas9 dissociating from the DNA product. We demonstrate that Cas9 turnover is limited by retention of the PAM-containing DNA product and occludes binding of new targets. Together, this work answers a long-standing question regarding exceptionally slow Cas9 turnover and offers a structural basis for Cas9 single-turnover kinetics.

## Results

### sgRNA truncation promotes faster Cas9 turnover

Cas9 turnover requires disruption of sgRNA:TS base-pairs, re-winding of the PAM-distal DNA and concurrent R-loop collapse. However, this base-pairing between the sgRNA and DNA target is required for conformational activation of the enzyme and promotes extremely stable association of Cas9, even following DNA cleavage. Eventually, Cas9 can be displaced by molecular motors when an enzyme collides with the PAM-distal side of the Cas9-bound DSB[27,28,31]. Interestingly, mutations in high-fidelity Cas9 variants that remove REC3 contacts with the PAM-distal end result in higher Cas9 turnover with on-target DNA via increased DNA re-winding but exhibit lower overall cleavage rates than WT Cas9[32,33]. We reasoned that shortening the RNA:DNA hybrid would result in faster DNA rehybridization, R-loop collapse and subsequent turnover.

We and other groups have shown that sgRNA truncation down to a 14-nt spacer can support cleavage, albeit at much slower rates[34-38]. To test whether truncated sgRNAs promote multiple-turnover kinetics, we first benchmarked Cas9 cleavage efficiency of a fluorescently labeled 55-bp target when used with a 15-nt and 17-nt spacer. When compared to the 17-nt sgRNA, we observed that cleavage with the 15-nt sgRNA was greatly inhibited (Supplementary Fig. 1a). Even after 20 h of incubation, we observed only a small fraction of the DNA substrate was cleaved.

DNA substrate topology and PAM-distal unwinding has been shown to modulate both cleavage efficiency with truncated sgRNAs and Cas9 displacement following DNA cleavage[39-41]. Given that cleavage with a 14-nt sgRNA was only observed when plasmids were used, we hypothesized that the truncated sgRNAs used in this study would perform better with plasmid DNA and may be a better substrate for testing turnover activity[38]. We then performed kinetic measurements of plasmid targets in 5-fold molar excess of Cas9 (Fig. 1a, b). The full-length 20-nt sgRNA supported rapid cleavage of plasmid DNA but quickly exhibited a plateau of linear product corresponding to a stoichiometric amount of Cas9, confirming single-turnover kinetics behavior as previously reported for *S. pyogenes* Cas9 (Fig. 1c)[3,32,42]. In contrast, cleavage with the 15-nt sgRNA was much slower but continued without exhibiting a clear plateau and exhibited ~2-fold increase

in turnover (Fig. 1c). This finding suggests that a shorter R-loop will be more likely to collapse following cleavage but is limited by the efficiency of catalytic activation.

We then explored how different guide-target combinations may retain turnover activity but increase cleavage rate. Extending the length of the truncated sgRNA from 15-nt to 17-nt greatly increases the rate of cleavage (Supplementary Fig. 1), and it has been established that Cas9 can tolerate mismatches in this region[20,43-48]. We hypothesized that extending the R-loop length while destabilizing it with mismatches could increase the rate of DNA rehybridization in the PAM-distal region and in turn, enhance turnover. To this end, we added one (1 mm) or two (2 mm) terminal mismatched nucleotides to the 5' end of the 15-nt sgRNA and tested turnover activity. The sgRNA with one mismatch (1 mm) contained 15 matched nucleotides with one additional mismatched nucleotide in position +16. The sgRNA with two mismatches (2 mm) contained 15 matched nucleotides with two additional mismatched nucleotides in positions +16 and +17. We hypothesized that if these mismatches were tolerated, these sgRNAs would exhibit similar cleavage activity to fully-hybridized 16-nt or 17-nt sgRNAs, but because the mismatches are less energetically stable than a fully-matched 17-nt sgRNA, we would increase the likelihood of DNA rehybridization post-cleavage and ultimately lead to faster R-loop collapse and release of product.

When compared to the fully hybridized 15-nt sgRNA, we observed enhanced turnover using both the 1 and 2 mm sgRNAs, with the highest being the 2 mm sgRNA (Supplementary Fig. 1). Excitingly, we found that the 2 mm sgRNA was able to support near-complete cleavage of the 5-fold excess plasmid substrate as well as rescued the cleavage defect observed for short, synthetic substrates as observed with the 15-nt sgRNA (Supplementary Fig. 1). To test whether this activity is consistent across other target sites, we designed two additional 2 mm sgRNAs targeting different sites in the lacZ gene of pUC19 (Supplementary Fig. 1) and performed the turnover assays as described above. Consistent with our previous results, both 2 mm sgRNAs targeting lacZ supported full cleavage of the target plasmid whereas the fully-matched 20-nt sgRNAs did not (Supplementary Fig. 1). Overall, we find that 2 mm sgRNAs promote multi-turnover activity across multiple plasmid targets.

### Product inhibition limits Cas9 turnover

The Cas9 reaction pathway is comprised of multiple intrinsic rate constants which can be challenging to define from observed rates[49,50]. Here, we use global fitting to define the cleavage and dissociation rate constants from our kinetic measurements by fitting the data to a model that accounts for all the data (Fig. 1). For the 20-nt sgRNA sample, we held the cleavage rate ($k_2$) constant at 60 min$^{-1}$ as it is too rapid to be defined by the measurements in our assay and has been extensively characterized in other studies (Fig. 1d)[20,25,26,38,51]. The observed rate of product release ($k_3$) with the 20-nt sgRNA was extremely slow (0.00085 min$^{-1}$) and consistent with single-turnover kinetics (Fig. 1e). For the 15-nt sgRNA sample, we observed the slowest cleavage rate of 0.109 min$^{-1}$ but with a dissociation rate (0.005 min$^{-1}$) approximately 10-fold faster product release than the full-length guide (Fig. 1d and e). We next determined the 2 mm sgRNA product release rate and find the fastest dissociation rate of 0.053 min$^{-1}$, 60-fold faster than a 20-nt sgRNA and therefore compensating for the reduced catalytic rate of 0.723 min$^{-1}$ (Fig. 1d and e).

Altogether, our kinetic analysis suggests that sgRNA truncation does promote product release following cleavage and that mismatches in the 5' region of the sgRNA:DNA hybrid favor rehybridization of the TS with the NTS post-catalysis, inducing spontaneous Cas9 dissociation from the product. As the reaction equilibrium shifts towards a higher concentration of the product relative to the substrate, there is a

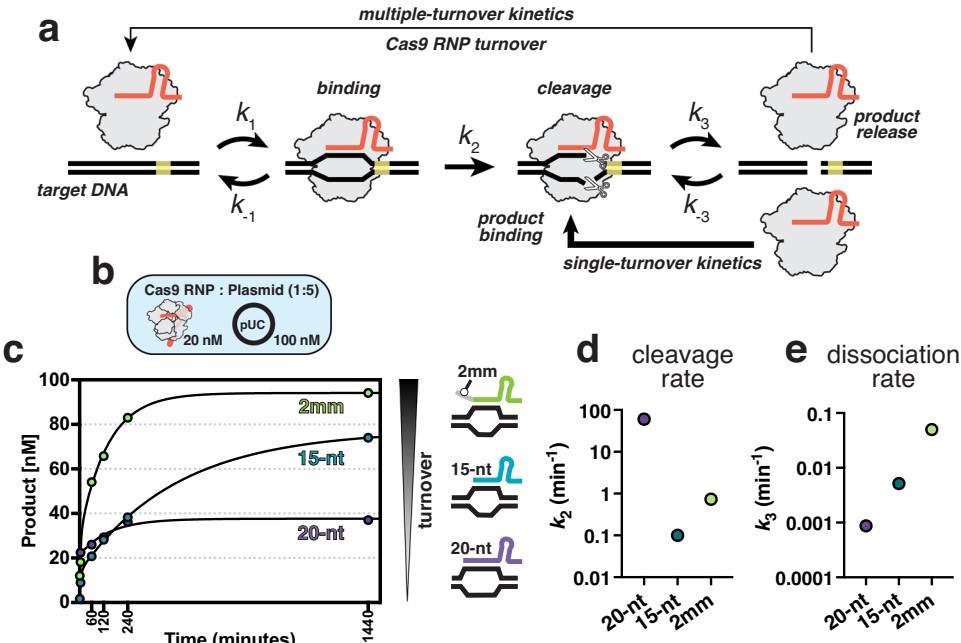

**Fig. 1 | Kinetic analysis of Cas9 programmed with truncated sgRNAs.**
**a** Schematic of Cas9 reaction cycle. **b** Diagram of the experimental setup for Cas9 turnover assays where plasmid DNA was added to a pre-formed Cas9 RNP in 5-fold molar excess. **c** Cas9 turnover assays with 20-nt, 15-nt, and 2 mm sgRNAs. 20 nM active Cas9 RNP was added to 100 nM plasmid substrate and product generation was plotted over time in minutes. **d** Cleavage rate constants ($k_2$) derived from global fitting of the data shown in panel **c**. The cleavage rate constant for the 20-nt sample was locked at 60 min$^{-1}$ as it was not captured in the time-scale used for the turnover assays and has been well-characterized in previous studies. **e** Dissociation rate constants ($k_3$) determined for the 20-nt, 15-nt, and 2 mm sgRNA samples. Source data are provided as a Source Data file.

higher population of product-bound Cas9, preventing binding of new substrates and slows down turnover as the reaction reaches completion (Supplementary Fig. 1).

## Kinetics-guided cryo-EM provides snapshots spanning the entire reaction cycle

We next sought to determine the molecular basis for Cas9 dissociation from DNA using cryo-electron microscopy (cryo-EM). Heterogeneity in cryo-EM datasets can be problematic when averaging particles to generate enough signal to produce a high-quality reconstruction because while averaging offers a net gain in resolution, high degrees of motion and heterogeneity leads to loss of information in regions that are often the most interesting. Accordingly, samples are typically prepared to maximize homogeneity to determine a single, high-resolution structure.

Cas9 is particularly susceptible to the heterogeneity problem as many domains undergo large structural changes during the reaction and are typically poorly resolved. To circumvent this, many studies have used catalytically inactive complexes or modified DNA substrates to trap Cas9 in specific conformations[18,21,52,53] but as these represent off-pathway intermediates, we decided to implement a different approach. This strategy involves using kinetics to guide cryo-EM sample preparation and has been previously used by our group to yield datasets capturing multiple on-pathway intermediate Cas9 structures under conditions that support catalytic activity[20,51]. By carefully selecting timepoints for vitrification, we enrich for maximal heterogeneity to capture rare, transient intermediates that are otherwise missed. As Cas9 dissociation is slow, this is an extremely rare state and structural details regarding dissociation following DNA cleavage remain uncharacterized. Faster dissociation kinetics observed with our 2 mm sgRNA sample made it the perfect candidate to visualize Cas9 in the process of dissociating from its product.

Based on our kinetic analysis with the 2 mm complex, we hypothesized that a timepoint mid-way through the reaction would capture

Cas9 during all stages of the reaction. To prepare samples for cryo-EM, we mixed Cas9 and the 2 mm sgRNA for 10 min at RT and initiated the reaction with addition of a 55-bp dsDNA target. This sample was incubated for 2 h at 37 °C before application to cryo-EM grids and subsequent vitrification. To classify the conformational states existing in our dataset, we used multiple rounds of 3D classification implemented in cryoSPARC[54] and generated an ensemble of reconstructions comprising a total of 25 subclasses with resolutions ranging from 2.8 to 3.4 Å. (Fig. 2b, Supplementary Fig. 4). Each subclass was manually inspected and sorted into five distinct conformational states (Fig. 2b, Supplementary Fig. 5). These states represent the progression of Cas9 through its reaction cycle and comprise the (I) pre-activation, (II) pre-cleavage, (III) checkpoint, (IV) product, and (V) dissociated states (Fig. 2c).

In the pre-activation state, Cas9 is bound to the target DNA where the R-loop has been fully formed but the PAM-distal DNA has not been fully accommodated into the central channel. In this state, HNH remains docked onto the RuvC domain and the DNA is in a linear conformation (Fig. 3a and d). Following R-loop completion, the DNA adopts a kinked conformation and induces large conformational changes in the nuclease domains. The pre-cleavage state represents this intermediate where we observe that the DNA is bent towards the central channel, diffuse density for both HNH and RuvC nuclease domains indicates rearrangement towards active conformations, and the DNA remains intact (Fig. 3a and d).

## The HNH domain resets following DNA cleavage

State III closely resembles the conformational checkpoint state (PDB 7Z4L, RMSD 0.796 Å) where the HNH domain has repositioned onto the TS:sgRNA heteroduplex but the scissile bond is still ~30 Å away from the active site (Supplementary Fig. 2). However, our structure has a critical difference where we observe the TS has been cleaved, evidenced by an unambiguous break in density along the TS backbone in our high-resolution map (Fig. 3b). This provides direct evidence that

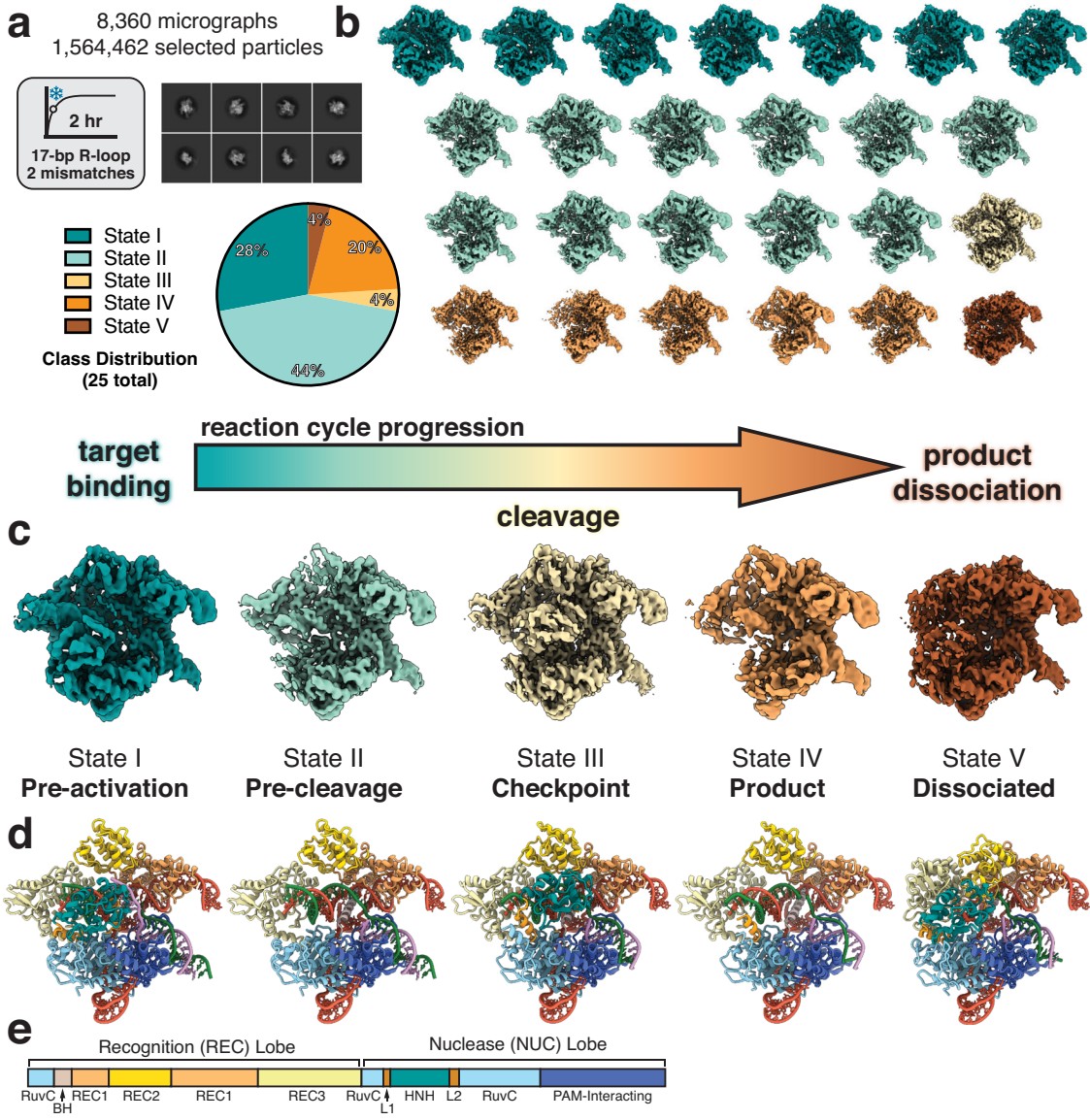

**Fig. 2 | Kinetics-guided cryo-EM data analysis. a** Data collection conditions used for cryo-EM. The pre-formed Cas9 RNP containing a 17-bp R-loop with 2 terminal mismatches was incubated with a 55-bp dsDNA target for 2 h at 37 °C before vitrification. Representative 2D classes are shown from the final particle stack used for further 3D classification. 3D class distribution represented as a pie chart. **b** Ensemble of cryo-EM reconstructions generated using 3D classification. **c** Representative maps corresponding to five main conformational states that span the entire reaction cycle from target binding to product dissociation. **d** Structures of each state corresponding to the maps shown in panel **c**. Colored according to the schematic below. **e** Cas9 architecture colored by domain.

following DNA cleavage, the HNH domain dissociates from the TS and resets to adopt the checkpoint conformation once again (Supplementary Fig. 2). In state IV, we see that Cas9 exists in the product state as well, but the nuclease domains become largely disordered and indicate that this is an intermediate following DNA cleavage where HNH remains highly flexible (Fig. 3b). A previous single-molecule FRET (smFRET) study has shown that following DNA cleavage, HNH adopts the conformational checkpoint state again[55], but was later confounded by a separate smFRET study showing HNH is highly flexible following cleavage[56]. Our findings reconcile both FRET studies by directly showing that after TS cleavage, the HNH domain does reset to the conformational checkpoint state and subsequently displays a high degree of flexibility. Given the timescale of these FRET studies (20 s) is much shorter than the lifetime of a Cas9 product complex (minutes to hours), our structural data provides further insight into longer-term dynamics of the HNH conformational landscape following DNA cleavage.

REC3 normally docks onto the 15–20 bp region of the distal duplex and imparts a kink in the DNA, facilitating the reorganization of the NUC lobe. In other product state structures, REC3 remains docked in this region[19–21]. In contrast, we find that in our product state structure, the REC3 domain has undocked from the PAM-distal duplex, moving away from the central binding channel and releasing the DNA kink to form a linear duplex once again (Supplementary Fig. 3). The PAM-distal duplex is also stabilized by electrostatic interactions within the RuvC domain, including a flexible, positively-charged loop that is normally only resolved in product state structures (Supplementary Fig. 3)[19]. In our product state, these residues are disordered and are not observed to establish the same contacts with the distal DNA backbone. Without stabilization by REC3 and RuvC contacts, the PAM-distal duplex exhibits a higher degree of flexibility following DNA cleavage and likely contributes to faster DNA rewinding and R-loop collapse. Interestingly, the only Cas9 homolog shown to be multi-turnover, *Staphylococcus aureus* Cas9 (SaCas9), does not retain these same

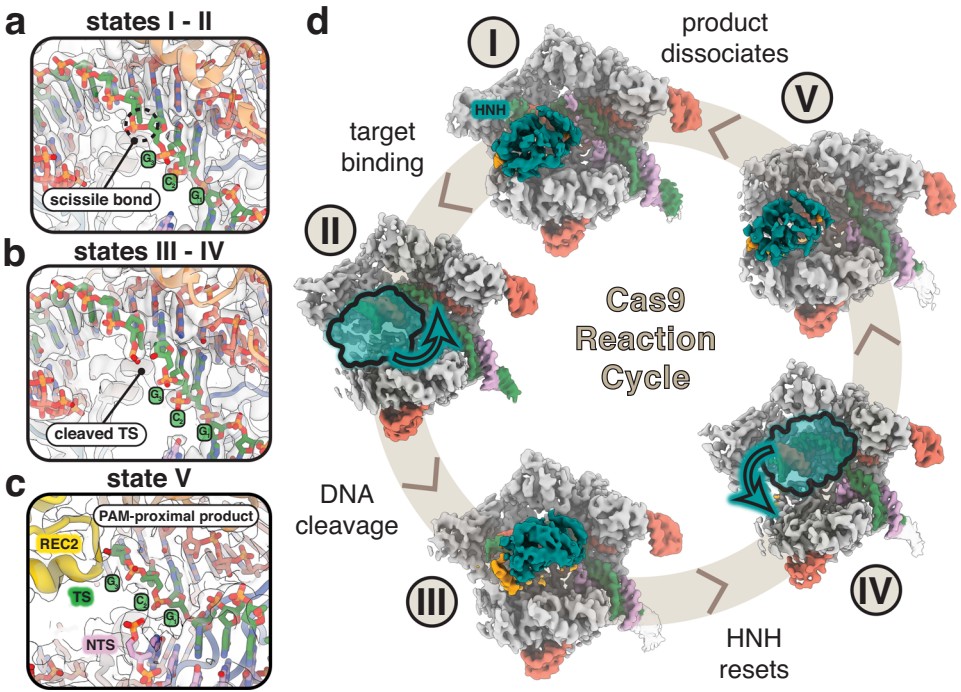

**Fig. 3 | Structures of Cas9 spanning the entire reaction from a single dataset.**
**a** Close-up view of the intact scissile bond of the target strand (TS) in states I and II. Map density shown from state II. **b** Close-up view of the cleaved scissile bond showing clear separation between nucleotides 3 and 4 of the hybridized TS. Map density shown from state III. **c** Close-up view of the cut site in state V following PAM-distal product release. PAM-proximal product remains bound where the first three nucleotides of the TS upstream of the PAM remain hybridized. **d** The five main conformational states populate the entire reaction cycle where (I) represents the

pre-activation state just following R-loop completion, (II) represents the pre-cleavage state during reorganization of the NUC lobe on-path to dock at the cleavage sites, (III) represents the conformational checkpoint state where HNH undocks following DNA cleavage, (IV) represents the product state where HNH remains flexibly tethered after undocking from the cut TS, and (V) represents the dissociated state where the PAM-distal end of the DNA is released from the complex.

interactions that stabilize the PAM-distal end of the duplex and this contributes to faster DNA re-winding following cleavage[32,42]. When comparing SpCas9 with SaCas9, SaCas9 has a smaller REC lobe, resulting in reduced surface area along the PAM-distal heteroduplex (Supplementary Fig. 3). These findings support a model where reduced interactions in the PAM-distal region leads to increased flexibility in the heteroduplex and promotes DNA rehybridization post-cleavage.

## Cas9 dissociates from the PAM-distal DNA product

Remarkably, in the dissociated state (state V), we observe particles in which the Cas9 complex has released the PAM-distal half of the DNA product and retains the PAM-proximal product (Fig. 4a and b). In this state, Cas9 reverts almost entirely back to its binary conformation where the REC2 and REC3 lobes move closer to the center of the complex and once again occlude the central DNA binding channel (Fig. 4b). While the sgRNA seed (1–10 nt) remains preordered for binding, the PAM-containing half of the product remains stably bound (Fig. 4c and d). Two residues, R1333 and R1335, in the PAM-interacting region directly participate in base-specific hydrogen bonding with the GG dinucleotides in the PAM and are maintained in the dissociated state (Fig. 4e). This persistent PAM-binding directly interferes with binding new targets and explains why Cas9 exhibits single-turnover kinetics.

Product dissociation and subsequent re-binding are difficult to deconvolute in vitro, especially when utilizing plasmid substrates. We hypothesized that this phenomenon could be more clearly observed with linear fragments given our observation of Cas9 dissociation in our structural dataset where we used short, linear

substrates. To assess PAM-distal and PAM-proximal product dissociation in vitro, we labeled the ends of a target DNA with different fluorophores and monitored product release with different sgRNAs after 2 h (Fig. 4f). When programmed with a 20-nt sgRNA, consistent with single-turnover kinetics, Cas9 tightly retains both ends of the DNA product. Strikingly, we observe significantly more product release of both DNA ends when Cas9 is programmed with the 2 mm sgRNA, exhibiting over a 50-fold increase in product release of the PAM-distal end of the duplex (Fig. 4f). In addition, we find that for all sgRNAs tested, there is higher ratio of PAM-distal product release over PAM-proximal product release (Fig. 4f). This supports a unifying model where Cas9 releases the PAM-distal end first, then subsequently releases the PAM-proximal end. In the case of higher turnover, the 2 mm sgRNA promotes faster release of the PAM-distal end and leads to a higher population of Cas9 bound to the PAM-proximal product. As the release of the PAM-proximal end is likely the same for any wild-type Cas9 RNP, retention of this PAM-proximal product remains the limiting factor governing whether Cas9 can target another site.

Our observations align with previous studies showing that the PAM-distal non-target strand (NTS) is released and accessible after cleavage[28], likely contributing to rehybridization with the PAM-distal TS and subsequent dissociation of the PAM-distal end of the DNA. Our findings reveal the structural and kinetic basis for slow turnover, provide direct visualization of Cas9 following product dissociation, and contribute to an updated model for the Cas9 reaction cycle (Fig. 5). Persistent binding of Cas9 to the PAM site emphasizes the importance of this interaction in the enzyme's turnover dynamics and

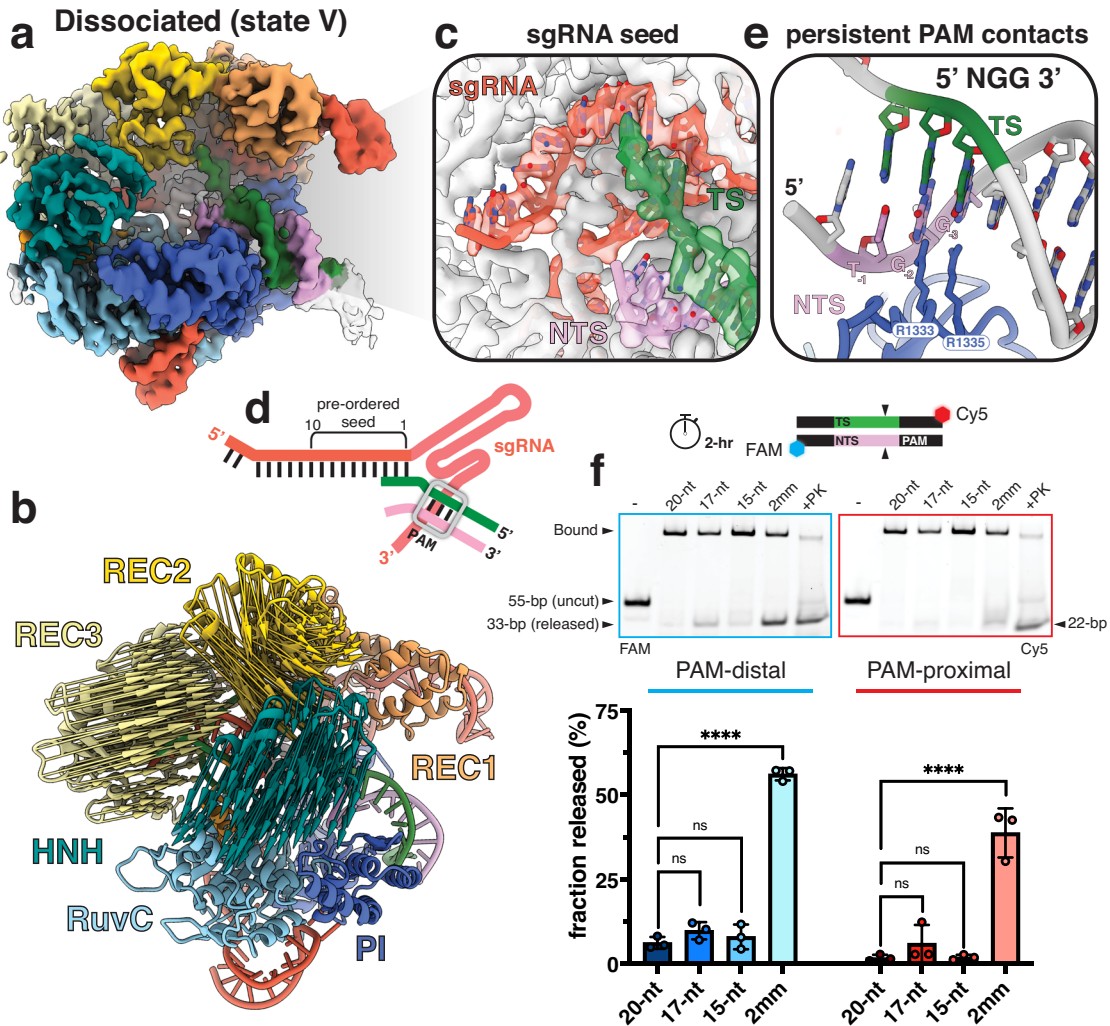

**Fig. 4 | Structural basis for slow Cas9 dissociation. a** Cryo-EM map of the dissociated state colored by domain. **b** Zoom-in of the pre-ordered sgRNA seed region. **c** Zoom-in view of the PAM contacts maintained even following PAM-distal product release. **d** Schematic of the nucleic acid in this complex. sgRNA colored in red, target strand in green, and non-target strand in pink. PAM site indicated with gray box. **e** Modevector arrows showing the domain movements from the checkpoint to dissociated states. **f** Monitoring PAM-distal and PAM-proximal product release following cleavage. Cas9 programmed with a 20-, 17-, 15-nt, or 2 mm sgRNA was incubated with target DNA labeled where the PAM-distal end was labeled with FAM and the PAM-proximal end labeled with Cy5. One sample containing 5X molar excess Cas9 20-nt RNP was included and treated with Proteinase K (+PK) as a control. Product release was analyzed at 2 h following reaction initiation on Native 4-20% polyacrylamide gels. Samples were quantified using ImageJ and plotted as a ratio of released product to bound product and represented as a percentage. Data represents the mean ± s.d. with $n = 3$. Statistical significance was determined using a one-way ANOVA with the Dunnett correction for multiple comparisons. ****$p = 2.74e{-8}$. Source data are provided as a Source Data file.

highlights a potential target for improving the efficiency of CRISPR-Cas9 gene editing.

## Discussion

This work contributes to our understanding of the entire Cas9 catalytic cycle and explains why Cas9 is a single-turnover enzyme on a structural level. We demonstrate that persistent binding of the PAM-containing DNA product inhibits Cas9 turnover and prevents binding to new targets. We discovered that decreasing the sgRNA spacer length and PAM-distal heteroduplex stability can promote R-loop collapse following DNA cleavage and transforms Cas9 into a multi-turnover enzyme. We leveraged our kinetic information to capture 25 high-resolution snapshots of Cas9 in action, including the first visualization of Cas9 dissociated from product DNA. By defining the mechanisms for Cas9 product release and subsequent turnover, our work solidifies a comprehensive, unifying model for the Cas9 reaction cycle (Fig. 5).

A multi-turnover Cas9 overcomes a significant bottleneck in the gene editing cycle and carries immense potential for biotechnology applications. Multi-turnover Cas9 complexes would no longer require cellular machinery to remove it from a DSB and subsequent repair would happen much faster. Persistent Cas9-DSB interactions can also modulate repair outcomes, skewing repair towards NHEJ[54]. HDR-mediated repair is limited by low efficiency and has been addressed by mutating or inhibiting genes involved in other DNA repair pathways[57,58]. It is plausible that a multi-turnover Cas9 would accelerate rates of HDR-mediated repair, circumventing the need to suppress other repair pathways and thus expanding upon available technologies for precision editing. Truncated sgRNAs have also been shown to increase editing fidelity in vivo[34,35]. Our findings that a truncated sgRNA with two mismatches enhances turnover emphasizes the importance of assessing off-target editing when employing truncated sgRNAs in vivo.

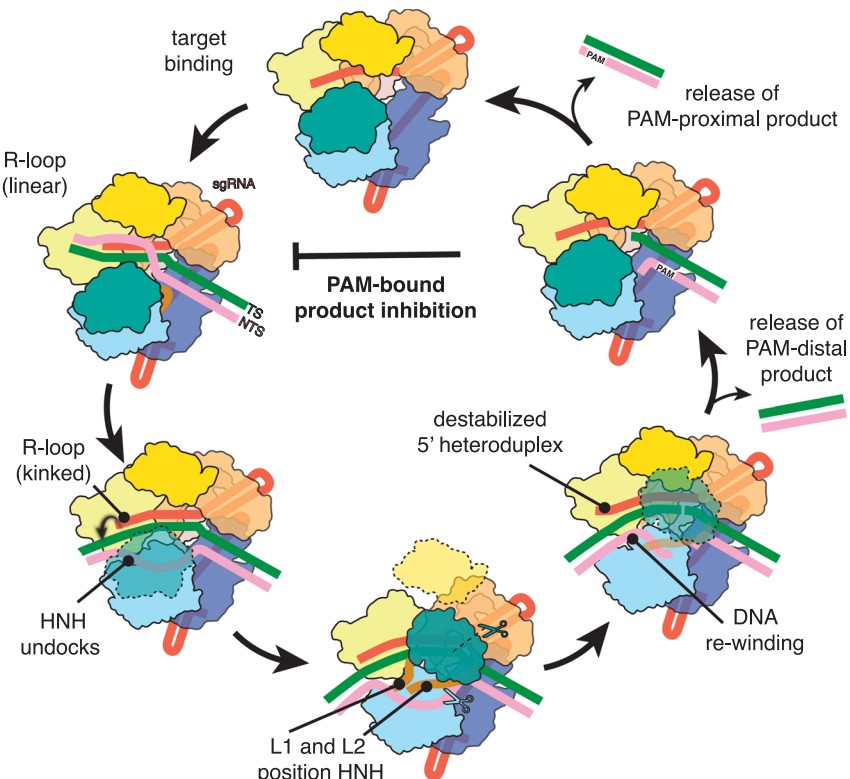

**Fig. 5 | Updated model of Cas9 reaction cycle.** The cycle begins with a Cas9-sgRNA binary complex searching for PAM-containing DNA targets. Complementarity of the sgRNA spacer and target sequence initiates R-loop propagation where the distal duplex adopts a linear conformation. When the R-loop is fully formed, the PAM-distal DNA duplex adopts a kinked conformation and initiates nuclease lobe reorganization. The L1 and L2 linker helices precisely position HNH onto the target strand. HNH reorganization exposes the RuvC active site and enables accommodation of the NTS. HNH and RuvC catalyze concerted TS and NTS cleavage, respectively. Following TS cleavage, the HNH domain moves in reverse and adopts the checkpoint conformation again. Coordinated cleavage of the NTS allows for PAM-distal DNA rewinding and triggers the slow release of the PAM-distal product. The PAM-proximal product remains bound to the complex, prevents rebinding of a new target, and leads to product inhibition. Eventual release of the PAM-proximal product resets Cas9 to the binary surveillance complex and restarts the cycle.

Many Cas9 re-engineering strategies have focused on designing Cas9 variants with expanded PAM recognition to increase the number of targets that are accessible for editing[59–62]. One PAM-less variant, SpRY-Cas9 (SpRY), was engineered to harbor 11 mutations in the PAM-interacting domain and can recognize all potential PAM sites[62]. Additional non-specific electrostatic interactions increase binding affinity for DNA in a sequence-independent manner and enables SpRY to interrogate many more sites in the genome, but this results in slow target search and accumulation at off-target sites[51]. Our finding that Cas9 remains tightly associated with the PAM, even after partial DNA release, suggest that SpRY may not only exhibit slow target search but likely retains even tighter binding to the product following cleavage, offering an additional explanation for its reduced efficiency. A similar strategy was recently employed to engineer two naturally high-fidelity Cas9 enzymes, PsCas9 and FnCas9[63,64]. Increasing Cas9 binding affinity to target DNA improved editing efficiency in vivo, but introduction of new electrostatic interactions with the PAM-proximal DNA may likewise impact turnover kinetics.

Overall, our study expands upon our current understanding of Cas9 biology and offers a novel strategy to turn Cas9 into a multi-turnover enzyme. We were able to interrogate the conformational landscape of Cas9 throughout the entire reaction cycle and uncover the elusive molecular mechanisms contributing to slow turnover of Cas9. This work provides critical insight that will guide both careful selection of the appropriate Cas9-based tool and the future design of new editors.

## Methods

### Cas9 expression and purification

*S. pyogenes* Cas9 (Cas9) constructs were cloned into a pET28b expression vector containing a C-terminal His$_6$ tag. Recombinant Cas9 was expressed in *Escherichia coli* strain OverExpress C41(DE3) (Sigma) and purified by Ni-NTA (nickel-nitrilotriacetic acid) and ion exchange chromatography (IEX). Expression was induced when cells reached an OD$_{600}$ of ~0.6 by adding isopropyl β-D-1-thiogalactopyranoside (IPTG) to a final concentration of 0.2 mM. Cells were incubated for 18–20 h at 18 °C while shaking and then pelleted at 4 °C by centrifugation at 7000×g for 30 min. The cell pellet was resuspended in lysis buffer (20 mM HEPES-NaOH, pH 7.5, 300 mM NaCl, 3 mM βME, 10% (v/v) glycerol) supplemented with a protease inhibitor tablet (Roche). Cells were sonicated on ice and the lysate was clarified at 18,000 × g for 45 min. Lysate supernatant was loaded onto a His-Trap FF crude column (Cytiva), washed with the lysis buffer containing 20 mM imidazole, and eluted with the same buffer but supplemented with 350 mM imidazole. Subsequent purification was performed using a HiTrap SP Sepharose High Performance cation exchange column (Cytiva) and a linear gradient of 100–500 mM KCl in 20 mM HEPES–KOH, pH 7.5. Fractions containing Cas9 were pooled, concentrated to 10 mg mL$^{-1}$, and buffer exchanged into Cas9 protein storage buffer (20 mM HEPES, pH 7.5, 200 mM NaCl, 1 mM EDTA, 10% (v/v) glycerol, 0.5 mM TCEP). All purified samples were aliquoted into single-use volumes, flash-frozen and stored at −80 °C.

## Nucleic acid preparation

The 55-bp DNA substrates were prepared by annealing two complementary ssDNA oligonucleotides (IDT). Each oligo was resuspended in water to a final concentration of 100 μM. Both oligonucleotides were combined at a 1:1 molar ratio and annealed by heating to 95 °C for 5 min and slowly cooling to room temperature. All sgRNAs used in this study were purchased from GenScript, resuspended in nuclease-free water to a final concentration of 20 μM, flash-frozen in liquid nitrogen, and stored at −80 °C. Sequences for oligonucleotides and sgRNAs listed in Supplementary Table 2. Fluorescently labeled substrates were prepared by annealing a 5′ 6-FAM TS oligo with an unlabeled NTS oligo. For the plasmid cleavage assays, the same Cas9 target site that was used in the linear fragments was cloned into pUC19, prepared in bulk via the ZymoPURE II Plasmid Maxiprep kit, and concentrated with the Vacufuge Plus Vacuum Concentrator (Eppendorf).

## Cryo-EM sample preparation and data collection

Cas9 complex samples were prepared in complex buffer (20 mM HEPES, pH 7.5, 200 mM NaCl, 1.0 mM EDTA, 10 mM MgCl$_2$, 0.5 mM TCEP) in a 1:1.2:1.2 molar ratio of Cas9:sgRNA:DNA. WT SpCas9 and the sgRNA were combined and incubated for 10 min at room temperature before adding the 55-bp dsDNA substrate to initiate the reaction. The ternary complex was incubated for 2 h at 37 °C and the reaction was quenched via vitrification. 2.5 μL of the sample was applied to glow-discharged holey carbon grids (Quantifoil 1.2/1.3), blotted for 7 s with a blot force of 0, and rapidly plunged into liquid nitrogen-cooled ethane using an FEI Vitrobot MarkIV. Data were collected on an FEI Titan Krios cryo-electron microscope equipped with a K3 Summit direct electron detector (Gatan). Images were recorded with SerialEM with a pixel size of 0.8332 Å at 13.3 electrons/pixel/second for 6 s (80 frames) to give a total dose of 80 electrons/pixel.

## Cryo-EM data processing

All data was processed in cryoSPARC v4.0[54] and the overall workflow is included in Supplementary Fig. 4. CTF correction, motion correction and template-based particle picking were performed in real-time using cryoSPARC Live v4.0.0. Particles were subject to multiple rounds of 2D classification and junk particles were filtered using multi-class ab initio reconstruction and subsequent heterogenous refinement. The filtered particle stack was subject to multiple rounds of 3D classification (10 classes per round) using the PCA initialization mode with forced hard classification. One final round of non-uniform refinement[65] was conducted using per-particle defocus and global CTF optimization parameters to generate the final 3D reconstructions. Cryo-EM map quality and model refinement statistics are included in Supplementary Table 1 and Supplementary Fig. 6.

**Atomic model building and refinement.** To build the atomic models, structures 6O0Z, 7S4V, 7Z4L, 7S4V, and 4ZT0 were rigid body fit into the pre-activation, pre-cleavage, conformational checkpoint, product, and dissociated states, respectively. 3D models were manually adjusted and inspected in Coot v1.0[66] and ISOLDE v1.4[67], and structures were real-space refined in Phenix v1.19[68]. All figures were generated in ChimeraX v1.8[69]. Statistics for data collection and model refinement are reported in Supplementary Table 1. Modevector arrows between aligned models were generated using PDBarrows v.1.0 as described in Chaaban et al. [70].

## Cas9 cleavage assays

Cleavage reactions were assembled in 1X cleavage buffer (20 mM HEPES, pH 7.5, 100 mM KCl, 0.5 mM TCEP, 10 mM MgCl2, 5% (v/v) glycerol). To form the Cas9-RNP, recombinant Cas9 was mixed with pre-annealed sgRNA in a 1:1.5 ratio and incubated at room temperature for 15 min. dsDNA was added to initiate the reaction and incubated at 37 °C. For turnover assays, DNA was added in a 5:1 molar ratio

(dsDNA:Cas9-RNP). At each timepoint, an aliquot of the reaction was quenched by the addition of 500 mM EDTA and 20 μg Proteinase K (Thermo Scientific). Cas9 cleavage products using fluorescently labeled 55-bp linear fragments were resolved on a 15% (w/v) TBE-Urea (7 M) polyacrylamide gel and ran at 180 V for 45 min. Plasmid cleavage products were resolved on a 1% agarose gel in 1X TAE buffer ran at 90 V for 75 min and post-stained with GelRed (Milipore Sigma). All gels were imaged using the ChemiDoc MP (Bio-Rad). Quantification of cleaved products was determined by densitometry analysis in ImageJ[71] and calculated as (total linear product/linear + nicked + supercoiled DNA) *100. Uncropped gels are provided in Supplementary Information.

**Native gel electrophoresis.** 55-bp fluorescently labeled substrates were prepared by annealing a 5′ 6-FAM NTS oligo with a 5′ Cy5 TS oligo in a 1:1 molar ratio. Reactions were assembled in 1X cleavage buffer and the Cas9-RNP was formed as described for the cleavage assays. Labeled dsDNA was added to initiate the reaction in a 1:1 Cas9-RNP:dsDNA ratio. Reactions were incubated for 2 h at 37 °C and visualized on a 4–20% non-denaturing polyacrylamide gel (Bio-Rad) in 0.5X TBE buffer run at 120 V for 30 min. Gels were imaged using the ChemiDoc MP (Bio-Rad) and the quantification of the released product was calculated by densitometry analysis in ImageJ[71]. The fraction released was calculated as (product released/product released + DNA bound).

**Kinetic analysis.** Kinetic measurements of Cas9 cleavage reactions were subject to global fitting in KinTek Explorer[72]. Turnover experiments were fit to the reaction scheme in Fig. 1 where: E + S $\rightleftharpoons$ ES $\rightarrow$ EP $\rightleftharpoons$ E + P. Experimental details of reactant concentrations were input for each experiment. In fitting data by simulation, each experiment is modeled exactly as it was performed. For the second-order DNA binding step ($k_1$), the rate was not defined by the data, and the binding rate constant and the reverse rate constant were locked at values to give a $K_d$ for DNA binding of 1 nM, similar to other estimates for the equilibrium constant for Cas9 binding to DNA[20,25,33]. The cleavage rate constant for the 20-nt sample was also locked at 60 min$^{-1}$ as it was not captured in the time-scale used for the turnover assays and has been well-characterized in previous studies[20,25,33]. Confidence contour analysis was performed using the FitSpace[73] function. These confidence contour plots are calculated by systematically varying a single rate constant and holding it fixed at a particular value while refitting the data allowing all other rate constants to float. The confidence interval was defined based on a threshold in $\chi^2$ calculated from the $F$-distribution based on the number of data points and the number of variable parameters to give the 95% confidence limits. A threshold of 0.99 was used to estimate the upper and lower limits for each rate constant. Rate constant values, their associated upper and lower limits, and the contour analysis are reported in Supplementary Fig. 7.

To generate the fit curves for visualization purposes only in Fig. 1c, we used a double exponential equation:

$$Y = A * (1 - e^{-k1t}) + k_2 t \tag{1}$$

where $Y$ represents the concentration of the product, $A$ represents the amplitude, $k_1$ represents the observed catalysis rate $k_{fast}$ for the initial burst phase of the reaction and $k_2$ represents the observed catalysis rate $k_{slow}$ for the linear phase of the reaction.

## Reporting summary

Further information on research design is available in the Nature Portfolio Reporting Summary linked to this article.

## Data availability

The atomic coordinates have been deposited in the Worldwide Protein Data Bank (wwPDB) under the accession numbers 9EAK (pre-activation), 9EAL (pre-cleavage), 9ED9 (conformational checkpoint),

9EDA [https://doi.org/10.2210/pdb9ED9/pdb] (product), and 9EDB (dissociated). Cryo-EM maps have been deposited in the Electron Microscopy Data Bank (EMDB) under accession numbers EMD-47834 (pre-activation), EMD-47835 (pre-cleavage), EMD-47941 (conformational checkpoint), EMD-47942 (product), and EMD-47943 (dissociated). Source data are provided with this paper.

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

## Acknowledgements

We thank Dr. Kenneth Johnson for assistance with kinetic analysis and helpful discussion as well as Dr. Jack Bravo and members of the Taylor lab for insightful comments on the manuscript. Data were collected at the Sauer Structural Biology Laboratory at the University of Texas at Austin. This work was supported by a National Institutes of Health grant R35GM138348 (to D.W.T.). The content is solely the responsibility of the authors and does not necessarily represent the official views of the National Institutes of Health. Computational resources for this work were supported by the Welch Foundation grant F-1938 (to D.W.T.).

## Author contributions

K.A.K. conceptualized, performed, and analyzed all work, and wrote the manuscript. D.W.T. supervised the study and acquired funding.

## Competing interests

The authors declare no competing interests.
