## [Transparent Peer Review file · Nature Communications]

Visualization of a multi-turnover Cas9 after product release

Corresponding Author: Dr David Taylor

Version 0:

Reviewer comments:

Reviewer #1

(Remarks to the Author)

Cas9 is a CRISPR RNA-guided DNA endonuclease, and *S. pyogenes* Cas9 (SpCas9) has been widely used for applications such as genome editing. SpCas9 remains bound to target DNA after cleavage, functioning as a single-turnover enzyme. This slow dissociation kinetics of SpCas9 could affect the efficiency of Cas9-mediated genome editing. However, the mechanism by which SpCas9 dissociates from target DNA after cleavage remains poorly understood. In this study, Kiernan et al. determined cryo-EM structures of the SpCas9-guide RNA-target DNA complex in multiple states, providing the first snapshots of Cas9 dissociating from product DNA. The structures revealed that while the PAM-distal product dissociates from SpCas9 after cleavage, SpCas9 remains bound to the PAM-proximal product, explaining why SpCas9 functions as a single-turnover enzyme. Overall, this manuscript has the potential to advance our understanding of the action mechanism of CRISPR-Cas9. The authors should consider the following points to improve the manuscript.

Points:

P5: As Ref. 38 showed that *Staphylococcus aureus* Cas9 (SaCas9) is a multiple-turnover, rather than single-turnover, enzyme, this sentence should be revised. In addition, it would be informative to add a discussion about why SpCas9 is a single-turnover enzyme, while other Cas9s such as SaCas9 are multiple-turnover enzymes, based on their structural differences.

P6: Why do one or two mismatched nucleotides at the sgRNA 5' end enhance turnover? Some explanations would be helpful. In addition, the authors should state that 2mm is visible in the structures, and show the entire heteroduplex including 2mm in Figure 3.

Minor points:

P2: "While the most widely used CRISPR-Cas enzyme is the *S. pyogenes* Cas9 endonuclease (Cas9)" should be rewritten as: "While the most widely used CRISPR-Cas enzyme is the Cas9 endonuclease from *Streptococcus pyogenes* (referred to as Cas9 for simplicity)."

P6: Figures 1d and 1e are not cited and should be cited.

P7: It would be better to explicitly state that the authors mixed Cas9, mm2-sgRNA (with a 15-nt guide), and a target DNA, incubated for 2 hours at 37°C, and then prepared the grids.

P7: "mm2" should be "2mm". In Fig. 4f, this should be corrected.

P8: How did the authors calculate the RMSD value? Was this calculated from equivalent C-alpha atoms?

P8: As "TS" is already defined on page 3.

P8: Citing "Figure 3b" seems incorrect in line 241. This should be corrected.

P11: "Truncated sgRNAs have also been shown to increase editing fidelity in vivo." - Please add references.

P11: Refs. 49-52 – A paper about SpCas9-NG should be cited (10.1126/science.aas9129). It would be better not to cite Ref. 50, as follow-up studies reported that xCas9 is less active in human cells.

P12: “A similar strategy was recently employed to engineer two naturally high-fidelity Cas9 enzymes, PsCas9 and FnCas9” - Please add references.

P13: “+18” and “+95” should be “18” and “95”, consistent with other parts of the manuscript.

P14: EDF5 should also be cited on page 7.

P15: “E + S = ES = EP = E + P” – “=” should be “→”.

Some figure citations seem incorrect. For example, Figures 4b and 4d are cited after Figure 4e, and EDF3 is cited before EDF2. These should be fixed.

Reviewer #2

(Remarks to the Author)

Summary:

In this study, Kiernan and Taylor use structural biology and enzymology techniques to interrogate the mechanism of slow product release by the Cas9 enzyme. The authors improved Cas9 product dissociation rates by using truncated and mismatched guide RNAs, demonstrating that associations between guide RNA and template DNA is a contributing factor to slow product release. These observations are further supported by the cryo-EM results where Kiernan and Taylor capture different catalytic states of the Cas9 in complex with the modified guide RNAs and template DNA. From one cryo-EM experiment, the authors captured five well-resolved Cas9 conformations including a novel “dissociation state” of Cas9-sgRNA bound to the post-cleavage DNA product. Ultimately, this work further extends the current enzymatic model of Cas9-catalyzed DNA cleavage by capturing rare transition state structures that explain the slow product release of Cas9.

Comments and Critiques:

The manuscript is well written and provides useful insights into why Cas9 shows slow product release. However, there are several points that need clarification:

1. Line 114-115 “We directly demonstrate that Cas9 turnover is limited by retention of the PAM-containing DNA product and occludes binding of new targets”. Please provide evidence to demonstrate whether the PAM-distal product is retained post catalysis and not upon re-binding.

2. Line 303 “We discovered that decreasing the sgRNA spacer length and PAM-distal heteroduplex stability can promote R-loop collapse following DNA cleavage and transforms Cas9 into a multi-turnover enzyme.” Line 303. Please provide a justification for the usage of a 5-fold higher substrate concentration compared to enzyme, ie why not use a greater excess of substrate.

3. Paragraph from lines 311-322. The CRISPR/Cas9 system is not presently optimized for in vivo applications due to off-target effects. Using the guide RNAs presented in this manuscript are likely to further propagate off-target effects of Cas9. Please comment on potential off-target effects associated with using the guide RNAs presented in the manuscript.

4. Lines 237-239 “HNH domain dissociates from the target strand (TS) and resets to adopt the checkpoint conformation once again” and Line 240 “nuclease domains become largely disordered following release of the heteroduplex” Doudna and coworkers present a similar structure to State II with density for the HNH domain (10.1126/science.aad8282). Please comment on this structure and how it fits this proposed model.

5. There appears to be a potential inconsistency between this manuscript and earlier work in the field that reference the “checkpoint state”. Can the authors clarify this please.

6. Please address the rationale for selecting the 17 bp + 2 mm sgRNA for the structural investigations and 15-bp + 2 mm sgRNA selected for the kinetics?

7. Would it be possible to obtain a structure of Cas9 in complex with the 17nt (2mm) and the PAM-proximal DNA?

8. Extended Figure 3. There seems to be a difference in DNA length resolved between 6O0X and product state Cas9 structure presented in this paper. Please comment on this difference in DNA length and its relevance to the conclusions.

9. Line 143 “We hypothesized that these truncated sgRNAs would perform better with plasmid DNA and may be a better substrate for testing turnover activity.” Provide reference and relevance.

10. Some minor comments on Figure 1 and Extended Figure 1:
a. Replicate number not stated and no reporting of error
b. What specific equations were used to fit the data?

- c. k2 was derived from previous studies – were the assay conditions identical? Was plasmid used as a substrate from the previous assay?
- d. All kinetic values are apparent and should be abbreviated with subscript app
- e. No control of fully uncut dsDNA in Extended Figure 1c.
- f. Validate “greatly inhibited”, and “a small fraction was cleaved” with respect to cleavage in Extended Figure 1c with densitometry (Lines 137-139)

Sriram Subramaniam and Michael Rowley

Reviewer #3

(Remarks to the Author)

Reviewer #4

(Remarks to the Author)

Kiernan and Taylor report a kinetic and structural analysis of Cas9 turnover. Cas9 is known to remain tightly bound to its products following cleavage, resulting in product inhibition and slow turnover kinetics. Cas9-product binding affects DNA repair and genome editing outcomes, so it is important to understand how Cas9 is product inhibited and whether turnover could be improved. Here, the authors show that Cas9 turnover can be induced by using shorter sgRNA spacer sequences and introducing mismatches in the PAM-distal region. Using this type of sgRNA, they solved the cryo-EM structures of several intermediates along the Cas9 catalytic cycle, including a unique structure in which Cas9 has dissociated from the PAM-distal product but remains bound to the PAM-proximal product. Based on this observation, the authors conclude that Cas9 turnover is limited based on tight binding to the PAM-proximal product, which inhibits its ability to bind to new targets.

The structural and biochemical data in this manuscript are high quality, and the manuscript is well-written. The manuscript also reports some interesting findings related to Cas9 turnover and how this turnover can be improved by engineering sgRNAs. However, it is unclear whether the model presented by the authors applies to all sgRNAs, or only the engineered sgRNA that they used for the structural study. Some aspects of the kinetic model for Cas9 turnover and product inhibition are also unclear. Below, I detail the aspects of the manuscript that could be clarified by changes to the text or additional experiments.

Major concerns

1. The authors describe a model in which the PAM-distal product is released prior to the PAM-proximal product, suggesting that PAM-proximal product release limits Cas9 turnover. However, in Fig. 4f, it appears that PAM-distal and PAM-proximal products remain tightly associated to Cas9 when a fully matched, 20-nt sgRNA is used, similar to observations in previous studies. The authors only observed faster PAM-distal product release when using a shorter sgRNA with mismatched nucleotides in the PAM distal region (2mm sgRNA), as might be expected due to destabilization of the PAM-distal sgRNA-DNA duplex. It is unclear whether the authors' model would apply to a fully matched sgRNA. The authors should either provide further evidence that the model applies to other sgRNAs, or they should clarify throughout the text that the model only describes how turnover of Cas9 is affected by the 2mm sgRNA.
2. It is also unclear how inhibitory PAM-proximal product binding is to Cas9 function. The authors only observed slower release of the PAM-proximal product for the 2mm sgRNA, which is also the sgRNA for which they observed the fastest Cas9 turnover. While it is possible that Cas9-2mm turnover is rate-limited by the PAM-proximal product, this is not clear from the way the results are presented throughout the manuscript, which describes the 2mm sgRNA as inducing Cas9 turnover. Because slower PAM-proximal product release was only observed for the sgRNA where significant turnover was observed, the two main conclusions of the manuscript seem contradictory.
3. The authors should clarify whether their kinetic model describes a situation in which Cas9 remains tightly associated with its products following cleavage, or whether Cas9 dissociates and reassociates with its product. The kinetic models shown in Fig. 1a and Extended Data Fig. 1d imply that Cas9 may be product inhibited based on re-binding to the product following product release. Similarly, on lines 182-183, the phrase “as the reaction equilibrium shifts toward a higher concentration of the product relative to the substrate” implies that Cas9 is only product inhibited at relatively high product concentrations, which is what would occur if Cas9 released the product and then re-bound to the product. This seems unlikely to be the case, as it is well established that Cas9 product inhibition occurs due to slow rates of product dissociation.
4. The authors provide quantified data for Cas9 turnover using the 2mm sgRNA against a plasmid target (Fig. 1e), but not against short DNA substrates that were used for the structural studies. Does Cas9 display similar turnover kinetics for short DNA oligos? In Extended Data Fig. 1a, it is not clear that substantial turnover was observed for such a substrate.
5. In Fig. 4f, it does not appear that nearly 50% of the PAM-proximal DNA was released from Cas9 for the 2mm sgRNA, as is shown in the quantification on the bottom of the panel. Also please define what +PK (presumably proteinase K) is and which sgRNA sample this treatment was performed for.

Minor concerns

6. The first sentence of the Discussion section may be overstated. It has long been known that Cas9 turnover is limited based on slow release of products, so it is unclear that a long-standing question is answered in the current study. The authors could also include more discussion of how their results compare with previous kinetic studies of Cas9 turnover (e.g. references 29 and 38).

7. It would be helpful if the authors could describe that the PAM-proximal DNA product remains bound to Cas9 in the “dissociated state” earlier in the Results section. I initially misunderstood the dissociated state to be the fully dissociated, Cas9-sgRNA binary complex, which was not clarified until the fourth sentence of the paragraph that describes this state.

8. In Extended Data Fig. 1b, the description of the 1mm and 2mm sgRNAs is a bit confusing. It would be helpful to explicitly state that the sgRNAs contained 15 matching nucleotides and 1 or 2 additional mismatched nucleotides (assuming that my interpretation was correct).

9. In Extended Data Fig. 1c, the figure legend has different time points than the label on the figure.

10. Which substrate was used for the kinetic curves shown in Extended Data Fig. 1e?

11. Please provide the concentrations of DNA and Cas9 used in Extended Data Fig. 1a and 1b.

12. In Fig. 4, the authors could switch the panel labels so that the panels could be cited in order in the text. Fig. 4e could be Fig. 4b, Fig. 4b could be Fig. 4c, and Fig. 4c could be Fig. 4e.

13. Lines 206, 210 and Fig. 4f: “mm2” should be “2mm” to be consistent with the rest of the manuscript.

14. Line 68: Consider changing the word “degradation” to “cleavage”, since Cas9 does not degrade DNA targets on its own.

15. Line 80: When introducing conformational control of Cas9 activity, it would be appropriate to also include reference to PMID 26524520

16. Lines 81-84: The authors could add references to support these sentences. In addition to some of the references cited on line 80, the authors could consider adding PMID 35422516 and 24912165.

17. Line 105: “however, this” should be “however, there”

18. Line 121-122: The sentence describing the requirement for Cas9 turnover is unclear. The phrase “release of the heteroduplex” implies that Cas9 turnover involves release of both the DNA and the sgRNA, which is not what the authors describe as turnover elsewhere in the manuscript. In addition, it doesn’t seem like R-loop collapse occurs subsequent to re-winding of the PAM-distal DNA, but is instead concurrent with this event.

19. Line 266-267: Should this sentence cite Fig. 4a rather than 4e?

20. Line 272-274: Please add a figure citation for this sentence.

Version 1:

Reviewer comments:

Reviewer #1

(Remarks to the Author)

The authors have addressed my concerns, and I support the publication of this manuscript in Nature Communications.

Reviewer #2

(Remarks to the Author)

The authors have satisfactorily addressed the concerns from the first round of review.

Reviewer #3

(Remarks to the Author)

Reviewer #4

(Remarks to the Author)

The authors have done an excellent job of addressing my concerns. I have one remaining suggestion, but otherwise feel that

the manuscript is suitable for publication.

-In figure 2a, I still feel it is a bit confusing/misleading to show the time course for cleavage of a plasmid in the schematic for how the grids were prepared, since the time course for cleavage of the target used in the cryo-EM experiment likely looks quite different. This could be removed from the figure without significantly altering the meaning of the figure.

Author's response to Reviewers' comments:

We wish to thank the reviewers for their positive feedback and constructive comments and suggestions for our manuscript. We have responded to all the comments and incorporated the corresponding changes to the main text and figures.

We believe that the revised form of the manuscript addresses the comments from the reviewers, and incorporation of the suggestions and additional experiments has improved the quality and scope of the manuscript.

All original comments from the reviewers are colored in black, responses from authors are colored in blue, and direct changes made to the text are colored in red.

REVIEWER COMMENTS

Reviewer #1 (Remarks to the Author)

Cas9 is a CRISPR RNA-guided DNA endonuclease, and *S. pyogenes* Cas9 (SpCas9) has been widely used for applications such as genome editing. SpCas9 remains bound to target DNA after cleavage, functioning as a single-turnover enzyme. This slow dissociation kinetics of SpCas9 could affect the efficiency of Cas9-mediated genome editing. However, the mechanism by which SpCas9 dissociates from target DNA after cleavage remains poorly understood. In this study, Kiernan et al. determined cryo-EM structures of the SpCas9-guide RNA-target DNA complex in multiple states, providing the first snapshots of Cas9 dissociating from product DNA. The structures revealed that while the PAM-distal product dissociates from SpCas9 after cleavage, SpCas9 remains bound to the PAM-proximal product, explaining why SpCas9 functions as a single-turnover enzyme. Overall, this manuscript has the potential to advance our understanding of the action mechanism of CRISPR-Cas9. The authors should consider the following points to improve the manuscript.

We thank the reviewer for their helpful feedback and suggestions. We have incorporated the input to improve the revised manuscript including new biochemical experiments, structural comparisons, and clarifications in the text.

Points:

P5: As Ref. 38 showed that *Staphylococcus aureus* Cas9 (SaCas9) is a multiple-turnover, rather than single-turnover, enzyme, this sentence should be revised. In addition, it would be informative to add a discussion about why SpCas9 is a single-turnover enzyme, while other Cas9s such as SaCas9 are multiple-turnover enzymes, based on their structural differences.

We have clarified that we are referring to SpCas9 in the sentence where Ref 38 is cited:

Main Text P5 Lines 148-150:

“confirming single-turnover kinetics behavior as previously reported for S. pyogenes Cas9 (Figure 1c)^{3,29,38}.”

We have also included structural comparisons between SpCas9 and SaCas9 in Extended Data Figure 2 and added discussion on why SpCas9 may be single-turnover versus SaCas9.

Extended Data P3 Lines (41-47):

“(f) Schematic showing the R-loop and REC lobe footprint along the heteroduplex. Dashed line indicates cut site. (g) Staphylococcus aureus Cas9 (PDB 7VW3) shown as transparent surface with the REC lobe colored by domain. Buried surface area (SA) of the heteroduplex upstream of the cut site and the REC lobe (residues 74-447). (h) Streptococcus pyogenes Cas9 (PDB 5F9R) shown as transparent surface with the REC lobe colored by domain. Buried surface area (SA) of the heteroduplex upstream of the cut site and the REC lobe (residues 95-731).”

P10 Lines 287-294:

“Interestingly, the only Cas9 homolog shown to be multi-turnover, Staphylococcus aureus Cas9 (SaCas9), does not retain these same interactions that stabilize the PAM-distal end of the duplex and this contributes to faster DNA re-winding following cleavage^{32,42}. When comparing SpCas9 with SaCas9, SaCas9 has a smaller REC lobe, resulting in reduced surface area along the PAM-distal heteroduplex (Extended Data Figure 3). These findings support a model where reduced interactions in the PAM-distal region leads to increased flexibility in the heteroduplex and promotes DNA rehybridization post-cleavage.”

P6: Why do one or two mismatched nucleotides at the sgRNA 5' end enhance turnover? Some explanations would be helpful. In addition, the authors should state that 2mm is visible in the structures, and show the entire heteroduplex including 2mm in Figure 3.

We appreciate the reviewers concerns and have added more context on why mismatches in this region of the truncated heteroduplex may enhance turnover on P6. Truncated guides have been shown to retain activity down to 14-nt but at a very diminished rate due to the PAM-distal DNA remaining hybridized in positions past 15-nt (Kiernan et al 2025 PMID: 39657754). Extending the length of the truncated sgRNA from 15-nt to 17-nt greatly increases the rate of cleavage and it has been established that Cas9 can tolerate mismatches in this region. We hypothesized that if the mismatches were tolerated, this would effectively behave as a 16-nt or 17-nt sgRNA and increase the cleavage rate. But because the mismatches are less energetically stable than a fully-matched 17-nt sgRNA, we would increase the likelihood of DNA

rehybridization post-cleavage and ultimately lead to faster R-loop collapse and release of product.

In reference to the reviewer's suggestion that we should show the entire heteroduplex in Figure 3, unfortunately due to the high flexibility in this region of the heteroduplex, density is quite diffuse in our maps for base pairs in positions +15 to +17 and can only be observed when filtering the maps to lower resolution. For this reason, we did not model those base-pairs and could not draw conclusions about how the mismatches were tolerated on a structural level.

P6 Lines 164-169:

"We hypothesized that if these mismatches were tolerated, these sgRNAs would exhibit similar cleavage activity to fully-hybridized 16-nt or 17-nt sgRNAs, but because the mismatches are less energetically stable than a fully-matched 17-nt sgRNA, we would increase the likelihood of DNA rehybridization post-cleavage and ultimately lead to faster R-loop collapse and release of product."

Minor points:

P2: "While the most widely used CRISPR-Cas enzyme is the *S. pyogenes* Cas9 endonuclease (Cas9)" should be rewritten as: "While the most widely used CRISPR-Cas enzyme is the Cas9 endonuclease from *Streptococcus pyogenes* (referred to as Cas9 for simplicity)."

We have reworded the sentence in the text accordingly:

P2 Lines 36-38:

*"While the most widely used CRISPR-Cas enzyme is the Cas9 endonuclease from *Streptococcus pyogenes* (referred to as Cas9 for simplicity), it exhibits single-turnover enzyme kinetics which leads to long residence times on product DNA."*

P6: Figures 1d and 1e are not cited and should be cited.

We have cited Figures 1d and 1e in the main text on P6 Lines 172-170).

P7: It would be better to explicitly state that the authors mixed Cas9, mm2-sgRNA (with a 15-nt guide), and a target DNA, incubated for 2 hours at 37°C, and then prepared the grids.

We have added more explicit preparation details in the main text.

P7 Lines 232-236:

"To prepare samples for cryo-EM, we mixed Cas9 and the mm2 sgRNA for 10 minutes at RT and initiated the reaction with addition of a 55-bp dsDNA target. This sample was

incubated for 2 hours at 37°C before application to cryo-EM grids and subsequent vitrification.”

P7: "mm2" should be "2mm". In Fig. 4f, this should be corrected.

This has been corrected.

P8: How did the authors calculate the RMSD value? Was this calculated from equivalent C-alpha atoms?

The RMSD was calculated in ChimeraX across all equivalent C-alpha atoms. We have now added a sentence in the Extended Data Figure 2 for clarity.

Extended Data P3 Line 47:

“RMSD calculated in ChimeraX across all equivalent C-alpha atoms.”

P8: As “TS” is already defined on page 3.

This has been corrected.

P8: Citing “Figure 3b” seems incorrect in line 241. This should be corrected.

In Figure 3b, we show a close-up of the cleaved scissile bond that is consistent with the map in that region. We have added additional text clarifying what exactly we are citing in that figure panel.

P8 Line 260:

“...evidenced by an unambiguous break in density along the TS backbone in our high-resolution map (Figure 3b).”

P11: "Truncated sgRNAs have also been shown to increase editing fidelity in vivo." - Please add references.

This has been added with Ref 31 and 32.

P11: Refs. 49-52 – A paper about SpCas9-NG should be cited (10.1126/science.aas9129). It would be better not to cite Ref. 50, as follow-up studies reported that xCas9 is less active in human cells.

We have taken out the citation for xCas9 and added doi: 10.1126/science.aas9129 to the reference list.

P12: “A similar strategy was recently employed to engineer two naturally high-fidelity Cas9 enzymes, PsCas9 and FnCas9” - Please add references.

This has been included with Ref 54 and 55.

P13: “+18” and “+95” should be “18” and “95”, consistent with other parts of the manuscript.

This has been corrected.

P14: EDF5 should also be cited on page 7.

Cited this figure on page 8.

P15: “E + S = ES = EP = E + P” – “=” should be “→”.

We have corrected this in the main text.

P15 Line 485:

Some figure citations seem incorrect. For example, Figures 4b and 4d are cited after Figure 4e, and EDF3 is cited before EDF2. These should be fixed.

We have reordered our extended data figures to be cited in the correct order.

Reviewer #2 (Remarks to the Author):

Summary:

In this study, Kiernan and Taylor use structural biology and enzymology techniques to interrogate the mechanism of slow product release by the Cas9 enzyme. The authors improved Cas9 product dissociation rates by using truncated and mismatched guide RNAs, demonstrating that associations between guide RNA and template DNA is a contributing factor to slow product release. These observations are further supported by the cryo-EM results where Kiernan and Taylor capture different catalytic states of the Cas9 in complex with the modified guide RNAs and template DNA. From one cryo-EM experiment, the authors captured five well-resolved Cas9 conformations including a novel “dissociation state” of Cas9-sgRNA bound to the post-cleavage DNA product. Ultimately, this work further extends the current enzymatic model of Cas9-catalyzed DNA cleavage by capturing rare transition state structures that explain the slow product release of Cas9.

We thank the reviewers for their thoughtful feedback and comments and have addressed them below. We have performed additional experiments and added more structural comparisons and textual discussion.

Comments and Critiques:

The manuscript is well written and provides useful insights into why Cas9 shows slow product release. However, there are several points that need clarification:

1. Line 114-115 “We directly demonstrate that Cas9 turnover is limited by retention of the PAM-containing DNA product and occludes binding of new targets”. Please provide evidence to demonstrate whether the PAM-distal product is retained post catalysis and not upon re-binding.

Our data support a model where PAM-proximal product retention is more likely due to the following supporting findings:

1. We observe higher PAM-distal product release over PAM-proximal product release with all guides that we've tested.
 - a. While overall product release was extremely slow for the fully-matched 20-nt sgRNA, measured PAM-distal product release was still ~3.5-fold higher than PAM-proximal ($6.21 \pm 1.74\%$ PAM-distal product released vs $1.74 \pm 0.97\%$ PAM-proximal product released)
2. Previous studies have shown that Cas9 displacement from double-strand breaks is dependent on Cas9 orientation
 - a. Translocating enzymes, such as T7 RNAP, can only displace Cas9 from the PAM-distal end (Clarke, R et al. 2018 PMID: 29979968) and polymerase collisions only from the PAM-proximal end act as a transcriptional roadblock (Hall, PM et al. 2022 PMID: 36471058).
 - b. In addition, single-molecule studies with the *Staphylococcus aureus* Cas9 enzyme indicate that SaCas9 also releases the PAM-distal end of the DNA before final dissociation from the PAM-proximal product, further supporting our model (Zhang S et al 2020 PMID: 32790142).

2. Line 303 “We discovered that decreasing the sgRNA spacer length and PAM-distal heteroduplex stability can promote R-loop collapse following DNA cleavage and transforms Cas9 into a multi-turnover enzyme.” Line 303. Please provide a justification for the usage of a 5-fold higher substrate concentration compared to enzyme, ie why not use a greater excess of substrate.

We decided to use 5-fold higher substrate concentration as it provided sufficient dynamic range within the timescale of the assay to accurately determine the dissociation rates of all complexes tested. In addition, when we modeled what the kinetic curves would look like based on our model in Kintek, using a larger excess of substrate did not better constrain the calculated rate constants for dissociation rate.

3. Paragraph from lines 311-322. The CRISPR/Cas9 system is not presently optimized for *in vivo* applications due to off-target effects. Using the guide RNAs presented in this manuscript are likely to further propagate off-target effects of Cas9. Please comment on potential off-target effects associated with using the guide RNAs presented in the manuscript.

We agree that CRISPR-Cas9 off-target editing is problematic for *in vivo* applications and is suboptimal for therapeutic applications without rigorous characterization of each target and its potential off-target sites. There have been multiple studies showing that truncated sgRNAs can improve fidelity and reduce off-target editing across multiple sites in mammalian cells (Fu, Y et al 2014 PMID: 24463574 and Kiani, S et al 2015 PMID: 26344044). As we did not test these sgRNAs *in vivo*, it is also possible that other factors such as target sequence composition, target site dynamics on the genome, and slower overall cleavage of the 2mm sgRNAs would impact *in vivo* editing efficiency and turnover. As such, we have highlighted in our discussion (P11 lines 328-330) that our results indicate these guides should still be carefully examined for off-targeting editing when implemented for high-fidelity applications.

P12 lines 359-360:

“Our findings that a truncated sgRNA with two mismatches enhances turnover emphasizes the importance of assessing off-target editing when employing truncated sgRNAs in vivo.”

4. Lines 237-239 “HNH domain dissociates from the target strand (TS) and resets to adopt the checkpoint conformation once again” and Line 240 “nuclease domains become largely disordered following release of the heteroduplex” Doudna and coworkers present a similar structure to State II with density for the HNH domain (10.1126/science.aad8282). Please comment on this structure and how it fits this proposed model.

We agree with the reviewers that PDB 5F9R is very similar to our state III structure as well as the structure 7Z4L (Pacesa et al 2022 PMID: 36002571). The major difference between those two structures and our state III checkpoint structure is that we observe clear target strand cleavage and believe that this provides additional insight into HNH movement post-cleavage. These discussions have been included in our results section and we have clarified that the nuclease domains become disordered in the fourth conformational state.

Page 9 Lines 256-266:

“State III closely resembles the conformational checkpoint state (PDB 7Z4L, RMSD 0.796 Å) where the HNH domain has repositioned onto the TS:sgRNA heteroduplex but the scissile bond is still ~30 Å away from the active site (Extended Data Figure 2). However, our structure has a critical difference where we observe the TS has been cleaved, evidenced by an unambiguous break in density along the TS backbone in our

high-resolution map (Figure 3b). This provides direct evidence that following DNA cleavage, the HNH domain dissociates from the TS and resets to adopt the checkpoint conformation once again (Extended Data Figure 2). In state IV, we see that Cas9 exists in the product state as well, but the nuclease domains become largely disordered and indicate that this is an intermediate following DNA cleavage where HNH remains highly flexible (Figure 3b)."

5. There appears to be a potential inconsistency between this manuscript and earlier work in the field that reference the "checkpoint state". Can the authors clarify this please.

We appreciate the reviewer noticing this inconsistency. As the checkpoint state has been used to represent different conformational states across previous studies, we are specifically using the convention for the conformational checkpoint state observed in Jiang et al 2016 10.1126/science.aad8282 and Pacesa et al 2022 PMID: 36002571. This state represents the conformation in which the HNH domain has undocked from the RuvC domain but adopts a catalytically inactive state where its active site is still situated over 30 Å away from the TS. This structure is distinct from the checkpoint state observed in Zhu et al 2019 PMID: 31285607 where HNH is still docked onto RuvC and more closely resembles our 'inactive' state before HNH undocking.

6. Please address the rationale for selecting the 17 bp + 2 mm sgRNA for the structural investigations and 15-bp + 2 mm sgRNA selected for the kinetics?

We have used the same sgRNA for both kinetics and our structural studies and have further clarified this in our results section.

P6 Lines 161-164:

"The sgRNA with one mismatch (1mm) contained 15 matched nucleotides with one additional mismatched nucleotide in position +16. The sgRNA with two mismatches (2mm) contained 15 matched nucleotides with two additional mismatched nucleotides in positions +16 and +17."

7. Would it be possible to obtain a structure of Cas9 in complex with the 17nt (2mm) and the PAM-proximal DNA?

If we interpret this correctly, we believe that our Dissociated State V structure is exactly this.

8. Extended Figure 3. There seems to be a difference in DNA length resolved between 600X and product state Cas9 structure presented in this paper. Please comment on this difference in DNA length and its relevance to the conclusions.

The reviewer is correct, and there is less PAM-distal DNA resolved in our maps when compared to 6O0X. We are only able to confidently model 15-bp of the heteroduplex in this state, but we observe the overall path of the distal duplex when filtering the map to lower resolutions. This is visualized in Extended Data Figure 3. Less PAM-distal DNA visualized in our product state supports our conclusions that there is higher flexibility in this region, especially after cleavage. This results in fewer REC3 contacts when compared to 6O0X, and ultimately, contributes to the higher rate of PAM-distal DNA re-winding and release.

9. Line 143 “We hypothesized that these truncated sgRNAs would perform better with plasmid DNA and may be a better substrate for testing turnover activity.” Provide reference and relevance.

Truncated sgRNAs have been shown to have reduced cleavage activity and is exacerbated when short oligo substrates are used. This has been shown in previous studies and has been used to guide usage of these for editing purposes. A recent study (Kiernan et al 2025 PMID: 39657754) has also shown that this cleavage inhibition with truncated sgRNAs can be alleviated when plasmids are used instead of short dsDNA oligos, and as we were investigating truncated sgRNA activity and kinetics, we chose to test plasmid substrates based on those previous studies. This has been referenced in P5 lines 141-145 and we added a line expanding on this.

P5 lines 141-145:

“DNA substrate topology and PAM-distal unwinding has been shown to modulate both cleavage efficiency with truncated sgRNAs and Cas9 displacement following DNA cleavage^{39–41}. Given that cleavage with a 14-nt sgRNA was only observed when plasmids were used, we hypothesized that the truncated sgRNAs used in this study would perform better with plasmid DNA and may be a better substrate for testing turnover activity³⁸.”

10. Some minor comments on Figure 1 and Extended Figure 1:

a. Replicate number not stated and no reporting of error

The full time course assays were plotted as a single replicate in KinTek and we have reported the associated confidence contour analysis showing the change in chi2 as a function of values for individual rates derived from the modeling. The chi2 threshold corresponding to the 95% confidence interval was used for reporting upper and lower limits on parameters and included in a new Extended Data Figure 7 (cited in the methods section). We have additionally performed turnover assays with two other plasmid targets and find consistently that 2mm sgRNAs promote turnover. This is included in Extended Data Figure 1 and in the main text.

P6 Lines 176-182:

“To test whether this activity is consistent across other target sites, we designed two additional 2mm sgRNAs targeting different sites in the lacZ gene of pUC19 (Extended

Data Figure 1) and performed the turnover assays as described above. Consistent with our previous results, both 2mm sgRNAs targeting lacZ supported full cleavage of the target plasmid whereas the fully-matched 20-nt sgRNAs did not (Extended Data Figure 1). Overall, we find that 2mm sgRNAs promote multi-turnover activity across multiple plasmid targets.”

P16 Lines 494-502:

“Confidence contour analysis was performed using the FitSpace⁷³ function. These confidence contour plots are calculated by systematically varying a single rate constant and holding it fixed at a particular value while refitting the data allowing all other rate constants to float. The confidence interval was defined based on a threshold in χ^2 calculated from the F-distribution based on the number of data points and number of variable parameters to give the 95% confidence limits. A threshold of 0.99 was used to estimate the upper and lower limits for each rate constant. Rate constant values, their associated upper and lower limits, and the contour analysis is reported in Extended Data Figure 7.”

b. What specific equations were used to fit the data?

To generate the fit curves for visualization purposes in Figure 1c we used the double exponential equation below and have included this in the methods section:

$$Y = A * (1 - e^{-k_1 t}) + k_2 t$$

To perform the global fitting of our kinetic data from which the actual rate constants were derived, we used KinTek Explorer (Johnson, K 2009 PMID: 19897109) with the model defined as below:

c. k_2 was derived from previous studies – were the assay conditions identical? Was plasmid used as a substrate from the previous assay?

Because Cas9 single-turnover kinetics have been extensively characterized previously (Refs 2,20,25,26) and we were more interested in accurately determining dissociation rates with the Cas9 samples turning over, it was not critical to determine the cleavage rate with the single-turnover 20-nt sgRNA in our assays. As cleavage occurs very rapidly, this would have required earlier timepoints (below one minute) to define the rate and has already been done in a previous study (Kiernan et al 2025 PMID: 39657754). The assay conditions were identical to this study including the use of plasmid substrates and the same buffer composition, complex assembly, reaction initiation, incubation

temperature, quenching method, and active site concentration determination. The apparent cleavage rate for the 20-nt sgRNA sample with plasmid substrates (0.54 min^{-1}) is consistent with the rate we chose to use in our modeling (1 min^{-1}). When inputting this rate in KinTek between 0.54 min^{-1} and up to the apparent rate determined with short oligos (60 min^{-1}), we did not see any change in the confidence of fit for the other rate constants that were defined by our model. This supports the fact that our assays do not contain data points that define this rate constant in our model and is not necessary for confidently calculating the intrinsic rate constants we reported.

d. All kinetic values are apparent and should be abbreviated with subscript app

The kinetic values determined by global fitting in KinTek estimate the intrinsic rate constants. Unlike apparent rates determined by fitting single equations, fitting the data by simulation allows us to solve equations numerically and are restrained by a model that provides a set of equations to define each constant.

e. No control of fully uncut dsDNA in Extended Figure 1c.

We do use uncut dsDNA controls but did not show it in the cropped gel image used in Extended Figure 1c due to the nature of loading on the gel (the uncut control and the lanes shown in Extended Data Figure 1c are 14 lanes apart). We have shown this and annotated the uncropped gels in a new Extended Data Figure 8. We would like to state that our plasmids samples contain dimers (verified by whole plasmid sequencing) and therefore two equivalent cleavage sites for Cas9. We see dimer isoforms (supercoiled, nicked, and linear) and when quantifying for linear product have included densitometry for every band but only show linear and supercoiled bands in the cropped gels for simplicity.

f. Validate “greatly inhibited”, and “a small fraction was cleaved” with respect to cleavage in Extended Figure 1c with densitometry (Lines 137-139)

We are referring to cleavage data shown in Extended Data Figure 1a and have quantified this. This is provided in the source data file.

Sriram Subramaniam and Michael Rowley

Reviewer #3 (Remarks to the Author):

Reviewer #4 (Remarks to the Author)

Kiernan and Taylor report a kinetic and structural analysis of Cas9 turnover. Cas9 is known to remain tightly bound to its products following cleavage, resulting in product inhibition and slow turnover kinetics. Cas9-product binding affects DNA repair and genome editing outcomes, so it is important to understand how Cas9 is product inhibited and whether turnover could be improved. Here, the authors show that Cas9 turnover can be induced by using shorter sgRNA spacer sequences and introducing mismatches in the PAM-distal region. Using this type of sgRNA, they solved the cryo-EM structures of several intermediates along the Cas9 catalytic cycle, including a unique structure in which Cas9 has dissociated from the PAM-distal product but remains bound to the PAM-proximal product. Based on this observation, the authors conclude that Cas9 turnover is limited based on tight binding to the PAM-proximal product, which inhibits its ability to bind to new targets.

The structural and biochemical data in this manuscript are high quality, and the manuscript is well-written. The manuscript also reports some interesting findings related to Cas9 turnover and how this turnover can be improved by engineering sgRNAs. However, it is unclear whether the model presented by the authors applies to all sgRNAs, or only the engineered sgRNA that they used for the structural study. Some aspects of the kinetic model for Cas9 turnover and product inhibition are also unclear. Below, I detail the aspects of the manuscript that could be clarified by changes to the text or additional experiments.

We thank the reviewer for their feedback and valuable comments. We have provided additional analysis and textual changes accordingly to address the concerns and improve our manuscript.

Major concerns

1. The authors describe a model in which the PAM-distal product is released prior to the PAM-proximal product, suggesting that PAM-proximal product release limits Cas9 turnover. However, in Fig. 4f, it appears that PAM-distal and PAM-proximal products remain tightly associated to Cas9 when a fully matched, 20-nt sgRNA is used, similar to observations in previous studies. The authors only observed faster PAM-distal product release when using a shorter sgRNA with mismatched nucleotides in the PAM distal region (2mm sgRNA), as might be expected due to destabilization of the PAM-distal sgRNA-DNA duplex. It is unclear whether the authors' model would apply to a fully matched sgRNA. The authors should either provide further evidence that the model applies to other sgRNAs, or they should clarify throughout the text that the model only describes how turnover of Cas9 is affected by the 2mm sgRNA.

We appreciate the reviewer's question and agree that the fully-matched sgRNA tested does exhibit much slower release of both the PAM-distal and proximal products. When comparing the fraction of PAM-proximal vs PAM-distal product released, we do still see a general trend of more PAM-distal product release with all guides that we tested, albeit

much less pronounced as with the mm2 guide. With the fully-matched 20-nt sgRNA, PAM-distal product release was ~3.5-fold higher than PAM-proximal ($6.21 \pm 1.74\%$ PAM-distal product released vs $1.74 \pm 0.97\%$ PAM-proximal product released). These data and uncropped gels have been included in the source data and extended data files.

Previous studies have also shown that displacement of Cas9 from double-strand breaks occurs in a directional manner where either translocating enzymes like T7 RNAP can only displace Cas9 from the PAM-distal end (Clarke, R et al. 2018 PMID: 29979968) and polymerase collisions only from the PAM-proximal end act as a transcriptional roadblock (Hall, PM et al. 2022 PMID: 36471058). In addition, single-molecule studies with the *Staphylococcus aureus* Cas9 enzyme indicate that SaCas9 also releases the PAM-distal end of the DNA before final dissociation from the PAM-proximal product, further supporting our model (Zhang S et al 2020 PMID: 32790142).

2. It is also unclear how inhibitory PAM-proximal product binding is to Cas9 function. The authors only observed slower release of the PAM-proximal product for the 2mm sgRNA, which is also the sgRNA for which they observed the fastest Cas9 turnover. While it is possible that Cas9-2mm turnover is rate-limited by the PAM-proximal product, this is not clear from the way the results are presented throughout the manuscript, which describes the 2mm sgRNA as inducing Cas9 turnover. Because slower PAM-proximal product release was only observed for the sgRNA where significant turnover was observed, the two main conclusions of the manuscript seem contradictory.

We appreciate the reviewer bringing this discrepancy to our attention and have reworded some of our conclusions to better explain our hypotheses. We consistently observe higher PAM-distal product release over PAM-proximal product release with all guides that we've tested, which supports a model where Cas9 releases the PAM-distal end first, then subsequently releases the PAM-proximal end. It's possible that the structural rearrangements that occur following release of the PAM-distal product (it is a large rearrangement of many domains and visualized in figure 4b) are critical for destabilizing the PAM-proximal product enough to promote its subsequent dissociation. The rate of PAM-proximal end dissociation is likely similar for any Cas9 RNP that has released its PAM-distal product already but remains the limiting factor governing whether Cas9 can target any new target site.

P11 Lines 319-326:

"In addition, we find that for all sgRNAs tested, there is higher ratio of PAM-distal product release over PAM-proximal product release (Figure 4f). This supports a unifying model where Cas9 releases the PAM-distal end first, then subsequently releases the PAM-proximal end. In the case of higher turnover, the 2mm sgRNA promotes faster release of the PAM-distal end and leads to a higher population of Cas9 bound to the PAM-proximal product. As the release of the PAM-proximal end is likely the same for

any wild-type Cas9 RNP, retention of this PAM-proximal product remains the limiting factor governing whether Cas9 can target another site.”

3. The authors should clarify whether their kinetic model describes a situation in which Cas9 remains tightly associated with its products following cleavage, or whether Cas9 dissociates and reassociates with its product. The kinetic models shown in Fig. 1a and Extended Data Fig. 1d imply that Cas9 may be product inhibited based on re-binding to the product following product release. Similarly, on lines 182-183, the phrase “as the reaction equilibrium shifts toward a higher concentration of the product relative to the substrate” implies that Cas9 is only product inhibited at relatively high product concentrations, which is what would occur if Cas9 released the product and then re-bound to the product. This seems unlikely to be the case, as it is well established that Cas9 product inhibition occurs due to slow rates of product dissociation.

We appreciate the reviewers feedback and have clarified in the main text that we propose our model describes a situation where Cas9 remains tightly associated with the PAM-proximal product rather than releasing product and re-binding. We agree that it would be unlikely for Cas9 to release the PAM-proximal product and re-bind and only at very high concentrations of product and low abundance of new targets may the product be inhibitory to binding new substrates. We have changed Extended Data Figure 1f to have “persistent product binding” instead of product re-binding.

P7 lines 203-205:

“As the reaction equilibrium shifts towards a higher concentration of the product relative to the substrate, there is a higher population of product-bound Cas9, preventing binding of new substrates and slows down turnover as the reaction reaches completion (Extended Data Figure 1).”

4. The authors provide quantified data for Cas9 turnover using the 2mm sgRNA against a plasmid target (Fig. 1e), but not against short DNA substrates that were used for the structural studies. Does Cas9 display similar turnover kinetics for short DNA oligos? In Extended Data Fig. 1a, it is not clear that substantial turnover was observed for such a substrate.

We did not perform full time course experiments with short DNA oligos as the reviewer is correct that we did not observe complete cleavage of the substrate even after 20 hours of incubation. A recent study (Kiernan et al 2025 PMID: 39657754) shows that DNA cleavage with truncated sgRNAs is particularly attenuated when short linear substrates are used. This is due to the NTS blocking HNH docking and trapping these complexes in an inactive conformation until alleviated by further PAM-distal DNA unwinding. While we do see that the mm2 sgRNA cuts the short oligos better than the 15-nt sgRNA and releases more PAM-distal product than any other RNP we tested (Figure 4f), we don't observe turnover to the same degree as we do on plasmid substrates. It is possible that plasmid topology also plays a role in promoting the release

of the PAM-proximal product and enables faster turnover.

5. In Fig. 4f, it does not appear that nearly 50% of the PAM-proximal DNA was released from Cas9 for the 2mm sgRNA, as is shown in the quantification on the bottom of the panel. Also please define what +PK (presumably proteinase K) is and which sgRNA sample this treatment was performed for.

We agree that the band for the released PAM-proximal samples does not look like nearly 50% of released DNA. Densitometry analysis shows that between 30-40% of the DNA was released and this did not change when we analyzed the data again ensuring that only peaks corresponding to the released product bands were stringently selected. We have included the re-analyzed plot and will have the measured densitometry values and raw gels included in the source data and extended data files. We have specified in the Figure 4 legend that +PK is proteinase K treatment on a 20-nt sgRNA sample under saturating conditions (5X excess Cas9 RNP:dsDNA). We expected this to fully liberate the cleavage products but as Cas9 RNPs are particularly stable, some product still remained bound to intact Cas9. This sample was used only as a control to show the size of cleaved products on the native gel and was not used in quantification.

Minor concerns

6. The first sentence of the Discussion section may be overstated. It has long been known that Cas9 turnover is limited based on slow release of products, so it is unclear that a long-standing question is answered in the current study. The authors could also include more discussion of how their results compare with previous kinetic studies of Cas9 turnover (e.g. references 29 and 38).

We agree with the reviewer that referring to a long-standing question in that way may misrepresent what has been established regarding Cas9 turnover for many years. We have reworded the first sentence of the discussion accordingly.

P11 Lines 339-340:

“This work contributes to our understanding of the entire Cas9 catalytic cycle and explains why Cas9 is a single-turnover enzyme on a structural level.”

7. It would be helpful if the authors could describe that the PAM-proximal DNA product remains bound to Cas9 in the “dissociated state” earlier in the Results section. I initially misunderstood the dissociated state to be the fully dissociated, Cas9-sgRNA binary complex, which was not clarified until the fourth sentence of the paragraph that describes this state.

We have reworded the beginning of the results section describing this state to make it clear that only half of the DNA product has dissociated.

P10 Lines 297-300:

“Remarkably, in the dissociated state (state V), we observe particles in which the Cas9 complex has released the PAM-distal half of the DNA product and retains the PAM-proximal product (Figure 4a,b). In this state, Cas9 reverts almost entirely back to its binary conformation where...”

8. In Extended Data Fig. 1b, the description of the 1mm and 2mm sgRNAs is a bit confusing. It would be helpful to explicitly state that the sgRNAs contained 15 matching nucleotides and 1 or 2 additional mismatched nucleotides (assuming that my interpretation was correct).

The reviewer’s interpretation is correct, and we have clarified this further in the text to help explain the exact composition of the sgRNAs.

P6 Lines 161-164:

“The sgRNA with one mismatch (1mm) contained 15 matched nucleotides with one additional mismatched nucleotide in position +16. The sgRNA with two mismatches (2mm) contained 15 matched nucleotides with two additional mismatched nucleotides in positions +16 and +17.”

9. In Extended Data Fig. 1c, the figure legend has different time points than the label on the figure.

We have corrected this.

10. Which substrate was used for the kinetic curves shown in Extended Data Fig. 1e?

We used the plasmid substrate for generating these kinetic curves. The red curve is the actual data generated and shown in Figure 1. The black curve is simulated data based on the model without product inhibition and generated in Kintek explorer.

Extended Data Figure 1 Lines 35-38

“Kinetic curves of experimental data (2mm) from the plasmid cleavage assays displaying high degrees of product inhibition overlayed with simulated data generated in Kintek Explorer showing what the curve would look like without any product inhibition.”

11. Please provide the concentrations of DNA and Cas9 used in Extended Data Fig. 1a and 1b.

This has been added in the figure legend.

12. In Fig. 4, the authors could switch the panel labels so that the panels could be cited in order in the text. Fig. 4e could be Fig. 4b, Fig. 4b could be Fig. 4c, and Fig. 4c could be Fig. 4e.

This has been updated accordingly.

13. Lines 206, 210 and Fig. 4f: “mm2” should be “2mm” to be consistent with the rest of the manuscript.

This has been corrected.

14. Line 68: Consider changing the word “degradation” to “cleavage”, since Cas9 does not degrade DNA targets on its own.

This has been changed accordingly.

15. Line 80: When introducing conformational control of Cas9 activity, it would be appropriate to also include reference to PMID 26524520

This reference has been included.

16. Lines 81-84: The authors could add references to support these sentences. In addition to some of the references cited on line 80, the authors could consider adding PMID 35422516 and 24912165.

We have added both those citations and supported the following sentences with appropriate citations.

17. Line 105: “however, this” should be “however, there”

This has been corrected.

18. Line 121-122: The sentence describing the requirement for Cas9 turnover is unclear. The phrase “release of the heteroduplex” implies that Cas9 turnover involves release of both the DNA and the sgRNA, which is not what the authors describe as turnover elsewhere in the manuscript. In addition, it doesn’t seem like R-loop collapse occurs subsequent to re-winding of the PAM-distal DNA, but is instead concurrent with this event.

We agree it was not clear what we were describing and have reworded the first sentence.

P4 Line 122-123:

“Cas9 turnover requires disruption of sgRNA:TS base-pairs, re-winding of the PAM-distal DNA and concurrent R-loop collapse.”

19. Line 266-267: Should this sentence cite Fig. 4a rather than 4e?

Yes, we thank the reviewer for catching this and have corrected it.

20. Line 272-274: Please add a figure citation for this sentence.

This has been added.

Response to final review

Reviewer #4 (Remarks to the Author):

The authors have done an excellent job of addressing my concerns. I have one remaining suggestion, but otherwise feel that the manuscript is suitable for publication.

-In figure 2a, I still feel it is a bit confusing/misleading to show the time course for cleavage of a plasmid in the schematic for how the grids were prepared, since the time course for cleavage of the target used in the cryo-EM experiment likely looks quite different. This could be removed from the figure without significantly altering the meaning of the figure.

We disagree with the reviewer, and we find the schematic helpful for general readers. The reviewer themselves noted that it does not change much regarding inclusion/exclusion.